# Short-Term Exposure to Ciprofloxacin Reduces Proteoglycan Loss in Tendon Explants

**DOI:** 10.3390/genes13122210

**Published:** 2022-11-25

**Authors:** Stuart James, John Daffy, Jill Cook, Tom Samiric

**Affiliations:** 1Department of Microbiology, Anatomy, Physiology and Pharmacology, La Trobe University, Melbourne, VIC 3086, Australia; 2Department of Infectious Diseases, St Vincent’s Hospital, Melbourne, VIC 3065, Australia; 3Sports and Exercise Medicine Research Centre, La Trobe University, Melbourne, VIC 3086, Australia

**Keywords:** fluoroquinolones, ciprofloxacin, tendinopathy, proteoglycans, equine, recovery

## Abstract

Fluoroquinolone antibiotics are associated with increased risk of tendinopathy and tendon rupture, which can occur well after cessation of treatment. We have previously reported that the fluoroquinolone ciprofloxacin (CPX) reduced proteoglycan synthesis in equine tendon explants. This study aimed to determine the effects of CPX on proteoglycan catabolism and whether any observed effects are reversible. Equine superficial digital flexor tendon explant cultures were treated for 4 days with 1, 10, 100 or 300 µg/mL CPX followed by 8 days without CPX. The loss of [^35^S]-labelled proteoglycans and chemical pool of aggrecan and versican was studied as well as the gene expression levels of matrix-degrading enzymes responsible for proteoglycan catabolism. CPX suppressed [^35^S]-labelled proteoglycan and total aggrecan loss from the explants, although not in a dose-dependent manner, which coincided with downregulation of *m*RNA expression of MMP-9, -13, ADAMTS-4, -5. The suppressed loss of proteoglycans was reversed upon removal of the fluoroquinolone with concurrent recovery of MMP and ADAMTS *m*RNA expression, and downregulated TIMP-2 and upregulated TIMP-1 expression. No changes in MMP-3 expression by CPX was observed at any stage. These findings suggest that CPX suppresses proteoglycan catabolism in tendon, and this is partially attributable to downregulation of matrix-degrading enzymes.

## 1. Introduction

Fluoroquinolones (FQ) are a synthetic class of antibiotic used to treat a wide variety of infections. They boast a broad spectrum of antimicrobial activity, rapid GI absorption, high oral bioavailability, and are generally well tolerated [1]. However, their use is associated with an increased risk of tendinopathy and rupture [2], particularly of the Achilles tendon (~90% of all cases [3,4]); a condition now commonly referred to as FQ-associated tendinopathy [5]. The risk is compounded in populations with an already elevated risk of developing tendinopathy such as those on concurrent systemic corticosteroid therapy [6], the elderly [7], and those with renal pathology [8,9]. Multiple FQs have been associated with development of tendinopathies however ciprofloxacin (CPX) has been implicated as one of the most common causative agents in the literature [3,7,10]. Interestingly, the time of symptom onset is variable, with a median latency period of between 6–14 days [11]. However, one report indicated symptoms as early as 2 h after initial treatment [4], while other reports have indicated tendinopathy symptoms occurring weeks to months after cessation of treatment [12]. The literature also indicates that nearly half of the reported cases of FQ-associated tendinopathy resulted in Achilles tendon rupture after discontinuation of FQ therapy [13,14]. These clinical findings suggest a lack of return to normal tendon homeostasis and a prolonged period of elevated risk of FQ-associated tendinopathy after cessation of FQ therapy.

The aetiological mechanisms of FQ-associated tendinopathy are still not fully understood. Toxicological studies have shown that FQs induce cytotoxic changes in various mammalian cell types [15] including tendon-derived cells [16,17,18] such as G2/M cell cycle arrest [19] and oxidative damage to mitochondria [20]. We have previously shown similar cytotoxic effects by CPX treatment in tendon explant cultures, and further demonstrated that reduced tenocyte viability continued 8 days following cessation of CPX treatment [21]. From a histological perspective, FQ-associated tendinopathy is associated with hypercellularity, collagen disruption and degeneration, and a reduction in proteoglycan synthesis [16,18,21,22,23,24,25,26,27]. Such changes may, in part, be due to changes in the expression of matrix metalloproteinase (MMP) and ADAMTS enzymes and/or tissue inhibitors of MMPs (TIMPs) that have been shown to have important roles in regulating tendon extracellular matrix homeostasis [28,29,30,31,32,33,34,35]. Indeed, CPX has been reported to increase the expression of such ECM-degrading enzymes [16,17,19,36], however data from these studies is equivocal and contradictory findings are evident. Moreover, these studies have primarily relied on monolayered cell culture models and have been mostly limited to discussing these effects in the context of collagen metabolism [37].

Whilst Collagen Type I is the most abundant protein in normal tendon, forming 60–80% dry weight [38], the extracellular proteoglycans (PGs) play an important role in tissue hydration and stability [39,40,41,42], extracellular modulation of signalling pathways [43,44,45], and regulation of collagen fibrillogenesis and fibril integrity [46,47,48]. Steady-state PG turnover is therefore a vital feature of tendon homeostasis and PG catabolism is achieved by members of the MMP and ADAMTS enzyme families. MMP-3 has the capacity to degrade various tendon PGs in vitro [49] and has shown specific activity against the small leucine-rich proteoglycan (SLRP) decorin in skin-derived fibroblast cell cultures [50] as well as the large PG aggrecan in cartilage [51,52]. MMP-9 has demonstrated catabolic activity against large PGs aggrecan [53] and versican [54] as well as the SLRP biglycan [55] in cartilage. MMP-13 has been shown to be specifically active against both aggrecan and versican at similar cleavage sites to aggrecanase [56,57] as well as the SLRPs with preferential catabolism of biglycan and fibromodulin in rat [31] and human cartilage [58]. The aggrecanases ADAMTS-4 and ADAMTS-5 have demonstrated the ability to mediate degradation of aggrecan in cartilage and tendon in vitro [32,59,60,61], as well as cleavage of biglycan [30]. Additionally, ADAMTS-4 has been shown to cleave versican [62], decorin and fibromodulin [63].

As far as we are aware, research published to address the effect of FQs on the gene expression of ECM-degrading enzymes with specific focus on PG catabolism is lacking, and no study has investigated how FQs influence PG loss from tissue explants where tenocytes remain within their native ECM. With that in mind, this study used equine tendon explant cultures to investigate the effects of varying concentrations of CPX on the overall loss of newly synthesised PGs from the tissue, as well as concentration of catabolites of aggrecan and versican, and whether any observed effects are reversible in the absence of CPX. Additionally, to elucidate a potential mechanism of action of CPX on PG catabolism, the effects on the expression of ECM-degrading enzymes was also investigated.

## 2. Materials and Methods

### 2.1. Materials Used

Equine fetlocks were obtained from skeletally mature horses (1–3 years old) which had no known musculoskeletal issues at the time of killing and were provided by Tooradin Knackery (Tooradin, VIC, Australia). Ciprofloxacin lactate solution (100 mg/50 mL) was obtained from Sandoz (Sydney, NSW, Australia) for use in tissue explant culture. Low Glucose Dulbecco’s Modified Eagle’s Medium (DMEM), penicillin-streptomycin, and newborn calf serum (NBCS) were purchased from Life Technologies (Carlsbad, CA, USA). Sulfur-35 radionuclide and ULTIMA GOLD^®^ was purchased from PerkinElmer (Boston, MA, USA). PD-10 (Sephadex G-25) columns were obtained from GE Life Sciences (Uppsala, Sweden). Q-Sepharose Fast Flow anion exchange resin, Chondroitinase ABC, and Keratinase were purchased from Sigma-Aldrich (St. Louis, MI, USA). MilliporeSigma™ Amicon™ Ultra 2 centrifugal filter units were purchased from Thermo Fisher Scientific (Waltham, MA, USA). Mini-PROTEAN^®^ TGM^™^ precast 4–15% gradient gels, Immun-blot^®^ PVDF membranes, 4× Laemmli sample loading buffer, Precision Plus Protein™ dual colour protein standards, Clarity™ ECL Western blotting substrate, PureZOL^™^ RNA isolation reagent, Fatty and fibrous tissue RNA isolation kits (Cat#732-6820), and RNA-*c*DNA reverse transcription kits (Cat#170-8890) were purchased from Bio-Rad (Hercules, CA, USA). RNAlater^®^ was obtained from Qiagen (Hilden, Germany). All other reagents were of analytical grade.

### 2.2. Instrumentation and Software Used

Tri-Carb 1500 Liquid Scintillation counter was purchased from PerkinElmer (Boston, USA). Mini-PROTEAN tetra-cells, ChemiDoc™ XRS + Imaging System and C1000™ Thermal Cycler, and CFX96™ Real-Time system were purchased from Bio-Rad (Hercules, USA). Precellys 24 homogeniser and CK 28 ceramic homogeniser beads were purchased from Bertin Technologies (Aix-en-Provence, France). NanoDrop 2000 was purchased from Thermo Fisher Scientific (Waltham, MA, USA).

Prism 9.0 data analysis software was produced by GraphPad (La Jolla, California). ImageJ was developed by Image Lab™ Laboratory for Optical and Computational Instrumentation (University of Wisconsin, WI, USA). QuantaSmart software was developed by PerkinElmer. Nanodrop 2000 software was produced by Thermo Fisher Scientific. CFX Manager PCR software were produced by Bio-Rad. Premier Biosoft (Palo Alto, CA, USA) developed beacon primer design software.

### 2.3. Tendon Explant Cultures

Equine fetlocks were transported within one hour after killing to the laboratory space for processing and analysis. The midsubstance region of the superficial digital flexor tendon (SDFT) was dissected from equine fetlocks under aseptic conditions and cut into pieces of approximately 100 mg, being careful to remove the surrounding paratendon as this tissue has a greater cellularity, and a different ECM composition to tendon proper [64]. The tissue was incubated with [^35^S]-Sulphate (150 µCi/mL) in sulphate-free DMEM for 6h at 37 °C [32]. The tissue was distributed into individual vials and cultured for 4 days in DMEM containing 0, 1, 10, 100, or 300 µg/mL CPX. The lower range of concentrations was selected to include dosages that reflect serum CPX concentrations reported clinically, and higher dosages are consistent with similar in vitro toxicological studies [18,27,65,66]. The culture medium was changed every 2 days, and the collected medium was stored at −20 °C in the presence of proteinase inhibitors. To determine the radiolabelled PGs remaining in the tissue, at the end of the culture period, the tissue was extracted with 4 M Guanidine Hydrochloride (GnHCl; pH 6.1) in the presence of proteinase inhibitors (8 mg/L soybean trypsin inhibitor, 0.1 M aminocaproic acid, 0.01 M Ethylenediaminetetraacetic acid, 0.001 M benzamidine HCl, 0.002 M iodoacetic acid, 0.0001 M phenylmethylsulphonyl fluoride, 0.02% NaN_3_, 0.01% Triton X-100) at 4 °C for 72 h, followed by 0.5 M NaOH at room temperature for 24 h.

In other experiments investigating whether the effect of CPX on PG loss is reversible, after incubation with [^35^S]-Sulphate, explants were maintained for 4 days in DMEM with 0, 1, 10, 100, or 300 µg/mL CPX. The following day of culture, the medium of the explants was switched to DMEM alone for an additional 8 days.

In experiments investigating *m*RNA expression levels, at the end of the culture period, the explants were placed in RNA later for 24 h at 4 °C.

### 2.4. Measurement of Loss of Newly Synthesised Proteoglycans

Aliquots (0.5 mL) of the tissue extracts were applied to PD-10 (Sephadex G-25) columns, equilibrated, and eluted with 4 M GnHCl, 0.1 M sodium sulphate, 0.05 M sodium acetate, 0.1% (*v*/*v*) Triton X-100; pH 6.1. Aliquots of the eluted samples were then collected in vials containing a scintillant cocktail (ULTIMA GOLD^®^), and the radioactivity in each sample assessed by liquid scintillation counting.

### 2.5. Isolation and Analysis of Endogenous Proteoglycan Fragments

Proteoglycan extracts were concentrated by centrifuge filtration and were resuspended in 6 M urea, 50 mM sodium acetate buffer (pH 6.0) with 0.15 M sodium chloride. Samples were loaded on to 1 mL columns of Q-Sepharose anion-exchange resin equilibrated in the same buffer. Columns were washed with the same urea buffer containing 0.5 M sodium chloride, and PGs were subsequently eluted with the same buffer containing 0.65 M sodium chloride. Eluted PGs were concentrated by centrifugal filtration and resuspended in ~200 µL enzyme digestion buffer of Tris HCl containing 10 µL chondroitinase ABC, 10 µL keratinase and 20 µL proteinase inhibitors [67]. Samples were then deglycosylated at 37 °C with gentle agitation for 24 h. Digested samples were concentrated by centrifuge filtering and resuspended in ~500 mL dH_2_O.

Deglycosylated PG core fragment isolate concentration was measured using a BCA protein standard kit as per the manufacturer’s instructions (Thermo Fisher). Samples were then dissolved in sample loading buffer with 10% mercaptoethanol as a reducing agent to a total of 1 mL. The amount of sample dissolved in buffer was determined by measured protein concentration for each treatment, ensuring that the concentration of isolated PG was equal across all samples.

Samples were subjected to electrophoresis on 4–15% gradient SDS-PAGE and electro eluted on to PVDF membranes (300 mA, 4 °C, 90 min). Membranes were probed with monoclonal antibody BC-3 (Abcam ab3773-1; against N-terminal interglobular domain neoepitope ARGVIL), and polyclonal antibody VCAN (Abcam Ab19345; against N-terminal neoepitope DPEAAE).

Quantification of core proteins of aggrecan and versican were determined at day 4 of treatment and after 8 days of recovery via band densitometry using ImageJ [68] for control and all CPX concentrations.

### 2.6. Quantification of mRNA Expression for Matrix-Degrading Enzymes

*m*RNA expression of MMP-3, MMP-9, MMP-13, ADAMTS-4, ADAMTS-5, TIMP-1, TIMP-2, TIMP-3 was analysed in explants at the end of the treatment period, as well as at the end of the 8-day recovery period. Explants were placed in RNAlater at 4 °C overnight, and then stored at −80 °C for future analysis. Tissue was homogenised in PureZOL reagent using a Bertin Precellys 24 homogeniser with the aid of ceramic beads (CK 28) and a program consisting of 6000× *g* for 20 s. This was repeated five times until all visible tissue was dissolved. RNA was extracted from tissue samples using a Bio-Rad Fatty and Fibrous Tissue Kit, according to the manufacturer’s instructions. 

The procedure included digestion of genomic DNA with DNase I. RNA integrity and concentration were determined using a Nanodrop 2000. Complementary DNA (cDNA) was synthesised using a reverse transcription kit. 

The resulting cDNA was subjected to real-time PCR amplification in an iCycler iQ Detection System. Each sample was run in duplicate. The values obtained for *m*RNA expression for the genes of interest were normalised for GAPDH housekeeping gene in the same sample. This was calculated according to relative quantification method using the ΔΔCt method where Ct is the cycle number of the detection threshold, and ΔΔCt shows the difference in threshold cycle (ΔCt) between gene of interest and GAPDH. Equine oligonucleotide sequences of the specific primers used in this study were designed using Beacon Designer and are shown in Table 1.

### 2.7. Analysis of Data

Results are represented as mean ± standard error of the mean (SEM). Experiments were repeated on separate occasions using tendon tissue derived from a different horse each time. Each datum from an individual experiment is compared to a control sample from the same animal which had been analysed simultaneously. Overall experimental data are presented as a mean of all tissue samples compared to their respective control after a 4-day treatment or 4 days treatment followed by 8 days in DMEM alone (without treatment). All statistical tests were performed using Prism 9.0 software. Repeated measures one-way analysis of variance (ANOVA) with Greenhouse-Geisser correction followed by post hoc Dunnett’s-test was used to determine multiplicity adjusted significance between each treatment condition and control. Differences were considered statistically significant at *p* < 0.05.

## 3. Results

### 3.1. Newly Synthesised Proteoglycans

CPX inhibited radiolabelled proteoglycan loss from the matrix of tendon explants after 4 days treatment (Figure 1a). The levels of radiolabelled proteoglycans remaining in the matrix at the end of the treatment period increased from 80.3% (untreated group) to 85% and 86.7% at 1 µg/mL (*p* = 0.042) and 10 µg/mL (*p* = 0.044) CPX, respectively. Although the reduced loss was not shown to be dose-dependent, the trend was demonstrated in all treatment groups. 

Moreover, the reversibility of suppressed loss of radiolabelled proteoglycans by CPX was investigated. This involved a 4-day treatment period with CPX followed by an 8-day culture period in the absence of the FQ. The levels of radiolabelled proteoglycans remaining in the matrix at the end of the 8-day recovery period were similar in all treatment groups compared to the control group (~55–60% remaining in the matrix; *p* > 0.05; Figure 1b).

### 3.2. Endogenous Proteoglycan Fragments

Proteoglycan catabolites remaining in the tendon ECM were evaluated by Western blotting with monoclonal antibody BC-3 and polyclonal antibody VCAN (Figure 2). An aggrecan metabolite reacted with BC-3 in all untreated and treated groups with a molecular mass of ~70–75 kDa, which is consistent with detection of the aggrecanase generated N-terminal interglobular domain neoepitope ARGVIL previously detected in equine with this antibody [69]. 

The banding intensity was markedly reduced by ~30% in the explants treated 100 µg/mL CPX after 4 days compared to the untreated group (Figure 2a; *p* < 0.0001). However, after the 8-day recovery period, this banding intensity returned to similar levels to the untreated group (Figure 2b). 

No differences in the BC-3 banding intensity were observed in the other treatment groups after the treatment period. However, following the 8-day recovery period, a 35% increase in intensity of the BC-3-immunoreactive band was observed in cultures which were initially treated with 300 µg/mL CPX compared to control (Figure 2b; *p* = 0.013).

Similarly, versican metabolites reacted with VCAN in all untreated and treated groups ranging in molecular mass from 60–75 kDa. This is consistent with detection of ADAMTS-4-generated N-terminal neoepitope DPEAAE previously detected with this antibody in equine [69]. However, no significant differences in banding intensities were observed following treatment and/or 8-day recovery period between untreated and treated groups (Figure 2c,d; *p* > 0.05).

### 3.3. mRNA Expression of MMPs

The *m*RNA expression of MMP-3 following treatment and recovery periods was not significantly different between the untreated and treated groups (Figure 3a,b; *p* < 0.05). 

The *m*RNA expression of MMP-9 was significantly downregulated to 0.22-fold and 0.14-fold of control after 4 days treatment with 100 µg/mL (*p* = 0.011) and 300 µg/mL (*p* = 0.0006) CPX, respectively (Figure 3c). This downregulation remained in these treated cultures with MMP-9 expression at 0.54-fold and 0.48-fold of control at 100 µg/mL (*p* = 0.011) and 300 µg/mL (*p* = 0.036) CPX, respectively, after the 8-day recovery period (Figure 3d)

Furthermore, the *m*RNA expression of MMP-13 was significantly downregulated to 0.32-fold and 0.12-fold of control after 4 days treatment with 100 µg/mL (*p* = 0.048) and 300 µg/mL (*p* = 0.0007) CPX, respectively (Figure 3e). No significant difference in *m*RNA expression of MMP-13 was observed in cultures treated at lower concentrations of CPX (1 µg/mL and 10 µg/mL) after 4 days. Additionally, no significant difference in MMP-13 expression between treated and untreated explants was measured after the 8-day recovery period.

### 3.4. mRNA Expression of ADAMTSs

The *m*RNA expression of ADAMTS-4 was significantly downregulated to 0.49-fold of control after 4 days treatment with 100 µg/mL CPX (Figure 4a; *p* = 0.024). There was no difference in *m*RNA expression of ADAMTS-4 in the other cultures after the 4 day treatment period. Furthermore, no significant changes in ADAMTS-4 expression were measured in any of the treated groups after the 8-day recovery period (*p* > 0.05; Figure 4b).

The *m*RNA expression of ADAMTS-5 was significantly downregulated after 4 days after treatment with CPX to 0.39-fold of control at 1 µg/mL (*p* = 0.012), 0.67-fold of control at 10 µg/mL (*p* = 0.015), and 0.38-fold of control at 100 µg/mL (*p* = 0.0036; Figure 4c). There was no difference in *m*RNA expression of ADAMTS-5 in the cultures treated at the highest concentration at 300 µg/mL CPX. Additionally, no significant changes in ADAMTS-4 expression were measured in any of the treated groups after the 8-day recovery period (*p* > 0.05; Figure 4d).

### 3.5. mRNA Expression of TIMPs

The *m*RNA expression of TIMP-1, TIMP-2, and TIMP-3 were not significantly different between the untreated and treated groups following the 4-day treatment period (Figure 5). However, after the 8-day recovery period, there was a 1.5-fold upregulation of expression of TIMP-1 (*p* = 0.003) in the group treated with 300 µg/mL CPX (Figure 5b). In contrast, at the same time point, TIMP-2 was significantly downregulated in samples treated with 1 µg/mL CPX (*p* = 0.01; Figure 5d). No significant changes in TIMP-3 expression were measured following the recovery period (Figure 5f).

## 4. Discussion

Tendon pathology is a major side-effect in a small proportion of patients treated with FQs. Various studies have indicated changes in ECM turnover in FQ-treated patients and animal models [26,70,71,72,73]. To our knowledge, this study is the first to demonstrate that CPX can inhibit PG catabolism in equine tendon explants. This study showed that CPX inhibited the loss of ^35^S-labelled PGs by ~10%. These inhibitory effects were demonstrated using CPX concentrations well within the serum concentration range observed in humans clinically [74]. Furthermore, catabolism of PGs from the chemical pool was also shown to be inhibited as indicated by the significant reduction in aggrecan catabolites from explants as determined by Western blotting (shown in Figure 2). Aggrecan is the most predominant large PG in tendon ECM [32,75,76] and it has been demonstrated that the loss of aggrecan from tendon explants is mainly due to the activity of ADAMTS-4 and ADAMTS-5 [32,59,60,61]. MMP-3, -9 and -13 have also demonstrated catalytic activity against aggrecan in vitro [51,53,56,57], however with less efficiency than the ADAMTSs [52].

FQ-induced upregulation of IL-1B-potentiated MMP-3 expression by FQs has been reported in human tendon cell cultures [17] as well as canine cartilage explants [77]. Our results contradict these findings as we observed no significant effect of CPX on MMP-3 *m*RNA expression in equine SDFT explants after 4 days treatment. However, Siengdee, Pradit, Chomdej and Nganvongpanit [77] also reported MMP-9 downregulation by enrofloxacin (a metabolic precursor to CPX) and Corps, Harrall, Curry, Fenwick, Hazleman and Riley [17] observed downregulation of MMP-13, and our study corroborates these findings in whole tendon explants with both being downregulated by higher concentrations of CPX after 4 days treatment. The differences in molecular responses between this study and others may be due to the experimental systems used. Notably, the findings in this study may not reflect the same tenocyte response to FQs than in cell culture models with cytokine-induced inflammation [17,36,77]. To our knowledge, this study is the first to investigate ADAMTS *m*RNA expression in tendon after treatment with FQs. We observed a significant downregulation of both ADAMTS-4 after 4 days treatment with 100 µg/mL CPX, while ADAMTS-5 was downregulated by CPX at all concentrations except 300 µg/mL. Expression profiling has indicated that ADAMTS-4 and ADAMTS-5 are highly expressed in the ECM of articular tissues such as cartilage and tendon [78,79], suggesting that these ADAMTSs are highly active in turnover of PGs in the ECM of tendons. TIMP expression was not significantly affected by CPX which supports similar findings in rat tail tenocytes treated with CPX [19]. Menon et al. [80] reported CPX-induced upregulation of TIMP-1 *m*RNA while TIMP-2 *m*RNA remained unchanged in cultured human tenocytes. Our findings support the latter in a whole tissue explant model but cannot confirm the former in a whole explant model. 

Taken together, the results of the *m*RNA expression analyses in this study suggest that CPX significantly downregulates *m*RNA expression of enzymes associated with catabolism of PGs, with no concurrent reduction in TIMP *m*RNA expression, in whole tendon explants. It is likely that reduced PG loss within the ECM of the tissue may be attributed to the inhibition of ADAMTS-5 expression at the same concentrations observed in this study. For all enzymes except MMP-3, downregulation was observed at 100 µg/mL CPX, and this likely explains the observed reduction in cleaved aggrecan protein fragments from explants treated with 100 µg/mL CPX as these detected fragments are associated with ADAMTS cleavage in tendon [32] and both MMP-9 and MMP-13 have shown activity against aggrecan at the same loci in cartilage [53,56].

The mechanism by which CPX alters the expression of matrix proteinases remains unclear. The contrasting effects of FQs on different ECM-degrading enzymes in our study and others implies that FQs differentially affect transcription factors which influence MMP and ADAMTS gene promoter pathways such as NFkβ and AP-1 [81]. Indeed, various FQs have demonstrated the ability to inhibit transcription or activity of pro-inflammatory cytokines like IL-1β, IL-6, IL-8, and TNFα [82,83,84,85] and many of these factors are considered potent transcriptional activators of MMP [81] and ADAMTS [86] genes. FQs may also indirectly influence extracellular signalling factors that promote MMP transcription through increased PG synthesis. AP-1 and NFkβ signalling pathways are regulated by various MAPKs which can be activated by binding of extracellular growth factors [81,87], and SLRPs can sequester extracellular growth factors such as CTGF and EGF [43,44,45] which can supress associated signal transduction. Indeed, decorin has been demonstrated to suppress MAPK pathway activation in human retinal cells [88]. It is therefore plausible that the increased PG synthesis induced by CPX that we have previously demonstrated in the same model [21] results in elevated extracellular sequestration of signalling molecules and suppressed activation of associated signalling pathways which promote MMP and ADAMTS gene transcription. 

Another potential mechanism involved in altered MMP expression is FQ-induced ROS accumulation which has been reported in tenocytes and chondrocytes [20,25,73] as a causal relationship between intracellular ROS and induction of various MMP expression [89,90,91]. However, given that we exclusively observed downregulation of MMP and ADAMTS expression in this study and we did not investigate ROS accumulation, it is unclear what role oxidative stress played in our findings and further study is warranted.

There are similarities in our findings to what has been observed in overuse tendinopathies in humans such as downregulation of ADAMTS-5 *m*RNA [78]. However, the majority of our findings are in contrast to what has been observed in degenerative tendon associated with overuse; PG loss is significantly increased in pathological human patellar tendon with concurrent upregulation of MMP-9 expression [92]. Additionally, downregulation of MMP-3 [93,94] and TIMP-3 [78] expression is a commonly noted biochemical feature of degenerative tendinopathies as well as increased MMP-13 expression [95]. Considering the biochemical events that precede degenerative pathology in tendon are poorly understood [96], we propose that what we have observed in this study represents an early process that precedes degeneration on the pathological continuum. Imaging and immunohistochemical studies have demonstrated an apparent accumulation of glycosaminoglycans in overuse tendinopathy [97] which is proposed to induce tissue swelling and collagen fibre separation resulting in subsequent maladaptation to load and degenerative lesion formation characteristic of later stage tendinopathy [49]. The inhibition of proteoglycan loss from our explants treated with CPX at 1 µg/mL and 10 µg/mL may result in PG accumulation in tendon during clinical FQ therapy, particularly given that this effect was seen at a concentration below peak serum CPX concentration after a single intravenous dose [74]. It is unlikely that this directly induces degeneration per se but rather predisposes tendons to future injury or pathology upon loading. This would explain the low incidence of FQ-associated tendinopathy in the general population, and why risk is higher in patients who already have an elevated risk for tendon pathology such as the elderly, those receiving concomitant systemic corticosteroids, and renal failure patients. 

Our results further demonstrated that the inhibition of newly synthesised proteoglycan loss in explants treated with CPX returned to normal levels 8 days upon cessation of CPX treatment. It appears that the recovery of proteoglycan loss coincided with recovery of *m*RNA expression of most ECM-degrading enzymes to control levels during the same period with a concurrent downregulation of TIMP-2 *m*RNA at 1 µg/mL. This suggests that upon cessation of CPX treatment, tendon explants compensate for the reduced proteoglycan loss by increasing proteoglycan catabolism by mechanisms including, but not limited to, upregulation of catabolic enzyme expression as has been shown previously [98]. This would further explain the elevated levels of aggrecanase-generated proteoglycan fragments and TIMP-1 *m*RNA expression that we observed in explants treated with 300 µg/mL CPX following the 8-day recovery period. 

The reduced aggrecan fragment concentration in tissue extracts observed at 100 µg/mL CPX also recovered after 8 days removal from CPX and this coincided with recovery of ADAMTS 4 and 5 *m*RNA expression at this concentration. However, MMP-9 was still significantly downregulated compared to control after 8 days recovery in explants treated with 100 µg/mL and 300 µg/mL CPX, to a lesser degree than immediately after treatment. This occurred concurrently with an upregulation of TIMP-1 at 300 µg/mL. This suggests that while the tissue does seem to recover from the downregulation of matrix degrading *m*RNA expression, this recovery does not occur or is delayed for MMP-9. It is plausible that this delayed recovery could impair ECM PG metabolism in patients after they have completed a course of FQs, clinically. We have previously demonstrated that not only does cell viability in a similar equine SDFT model remain low following cessation of CPX treatment, but the viability of cells reduces even further after 8 days in DMEM alone [21]. This effectively rules out a rebound in viable cells as the mechanism for the recovery of the PG catabolic activity in explants observed in the current study. 

These findings provide further insight into the nature of PG metabolism in tendons after FQ therapy and may indicate a mechanism underpinning the delayed onset of FQ-associated tendinopathy reported clinically. 

This study has some limitations to consider when interpreting the discussed results. Firstly, we acknowledge the sample size in our study is relatively small; this was due to the number of fetlocks we could obtain being constrained by the operations of the donor Knackery. Additionally, the breed and sex of each equine used in this study was not disclosed by the donor Knackery at the time of tissue collection. Our method for measuring PG loss involves labelling PGs during a post-translational sulphation stage in the synthetic process. As such, our measurement of total PG loss from tendon explants represents only PGs which were synthesised during the [^35^S]-Sulphate incubation before treatment, as PGs synthesised before radiolabelling will have already been sulphated. Many of the discussed effects were surprisingly not dose dependent. We cannot explain this observation, and this should be explored in future studies. Moreover, many of the discussed effects occurred at the higher CPX concentrations used in this study (100 µg/mL or 300 µg/mL). Inclusion of these supraphysiological concentrations was based on consistency with other similar toxicological studies [18,27,65,99], however the highest concentration of FQs reported in articular tissue is 30 µg/mL [100] and peak concentrations achieved by CPX in tendon tissue are currently unknown [10]. As such, effects observed at these higher dosages may have limited clinical applicability.

## Figures and Tables

**Figure 1 genes-13-02210-f001:**
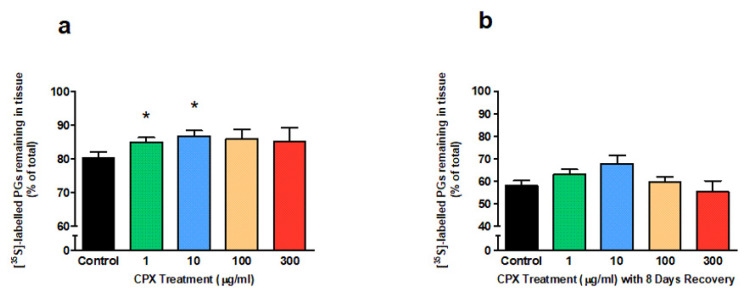
^35^S-sulphate labelled proteoglycans (PGs) remaining in equine derived tendon explant cultures after (**a**) 4 days treatment with 1, 10, 100 or 300 µg/mL CPX (n = 4) and (**b**) subsequent 8 days without CPX treatment (recovery period; n = 4). Values represent mean ± SEM and are expressed as % of total newly synthesised pool remaining in tissue. * *p* < 0.05 vs. control.

**Figure 2 genes-13-02210-f002:**
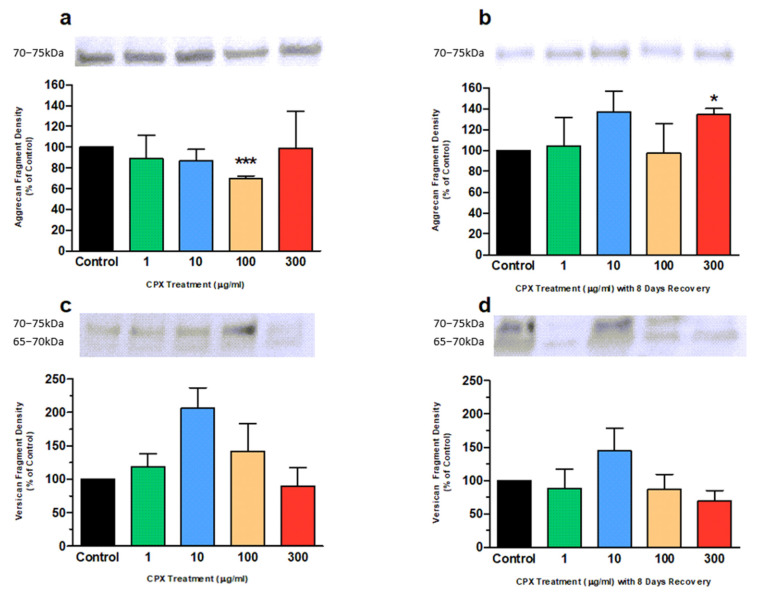
Densitometric analysis of the results of immunoblotting obtained from equine derived tendon explants after a 4 day culture period (**a**,**c**) and subsequent 8-day culture period without CPX treatment (**b**,**d**) with various concentrations of CPX as indicated. A monoclonal antibody (BC-3) against N-terminal interglobular domain neoepitope ARGVIL was used for detection of aggrecan fragments (**a**,**b**; n = 5) and a polyclonal antibody (VCAN) against N-terminal neoepitope DPEAAE was used for detection of versican fragments (**c**,**d**; n = 6). Values represent mean ± SEM and are expressed as % of control. ** p* < 0.05, *** *p* < 0.001 vs. control.

**Figure 3 genes-13-02210-f003:**
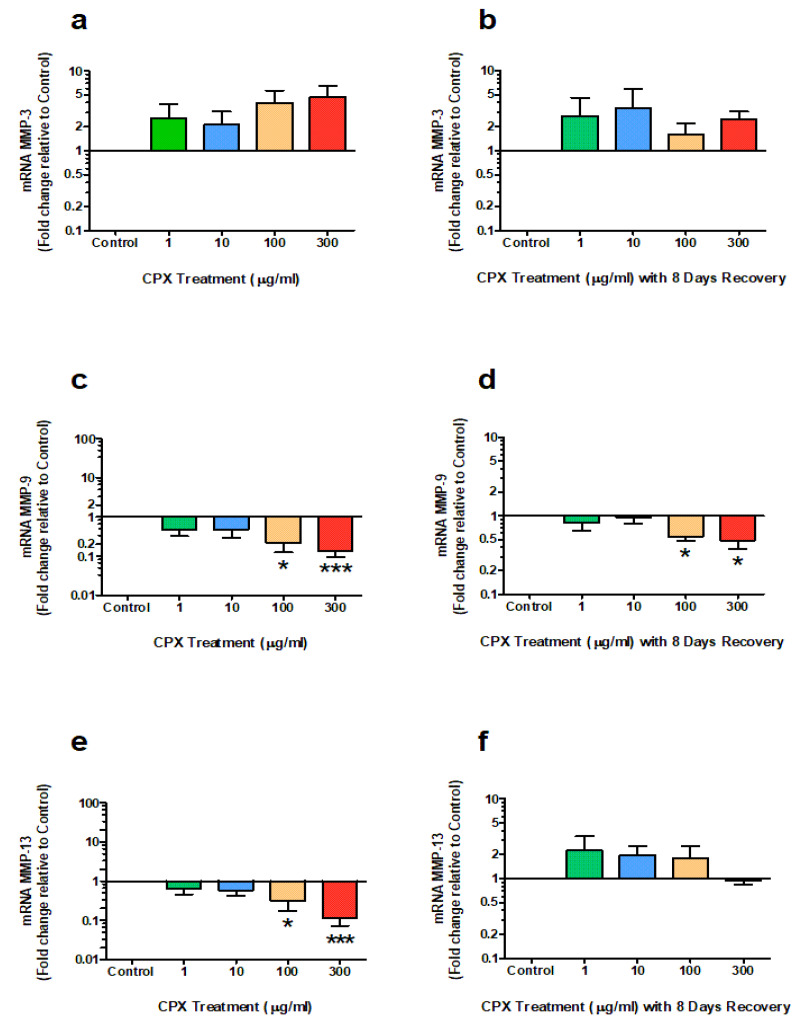
*m*RNA expression of MMP-3 (**a**,**b**), MMP-9 (**c**,**d**) and MMP-13 (**e**,**f**) after 4 days treatment with 1, 10, 100 & 300 µg/mL CPX and subsequent 8-day recovery period in equine derived tendon explant cultures (n = 4). Data are expressed logarithmically. Values represent mean ± SEM and are expressed as fold change relative to control, normalised to GAPDH. ** p* < 0.05, **** p* < 0.001 vs. control.

**Figure 4 genes-13-02210-f004:**
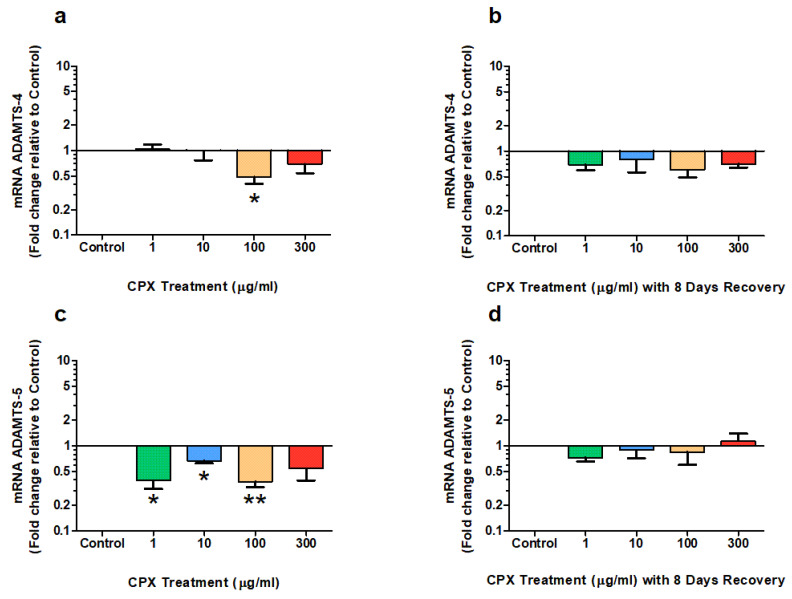
*m*RNA expression of ADAMTS-4 (**a**,**b**) and ADAMTS-5 (**c**,**d**) after 4 days treatment with 1, 10, 100 & 300 µg/mL CPX and subsequent 8-day recovery period in equine derived tendon explant cultures (n = 4). Results are expressed logarithmically. Values represent mean ± SEM and are expressed as fold change relative to control, normalised to GAPDH. ** p* < 0.05, *** p* < 0.01 vs. control.

**Figure 5 genes-13-02210-f005:**
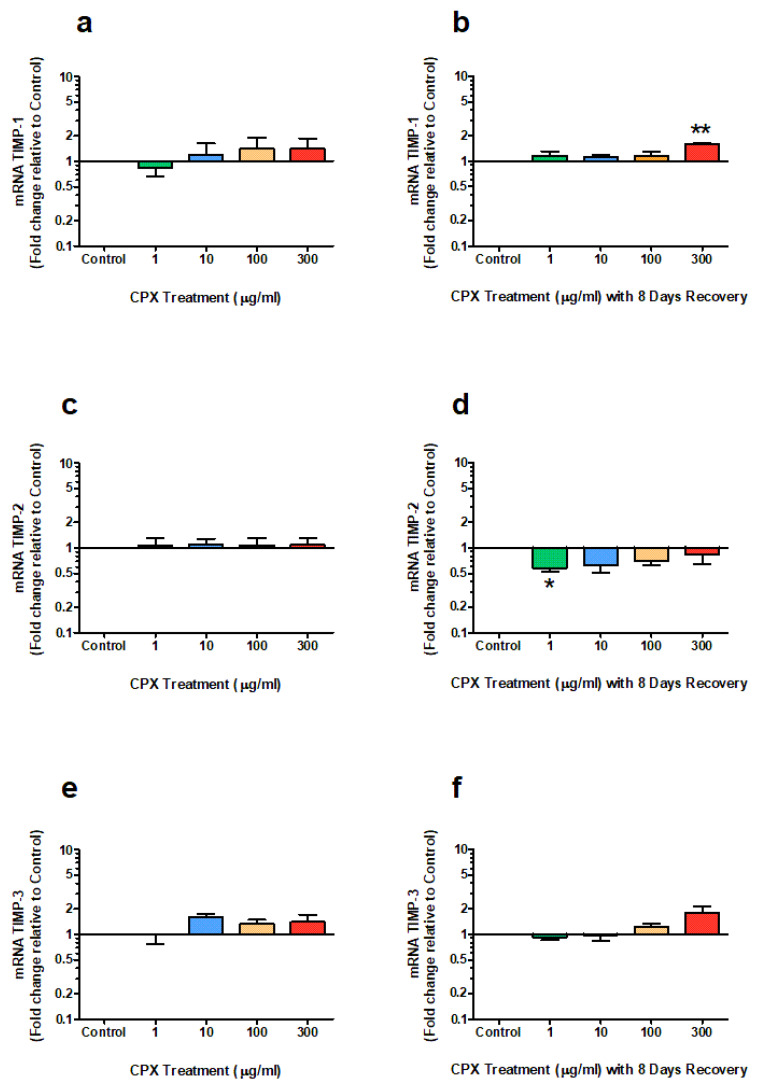
*m*RNA expression of TIMP-1 (**a**,**b**), TIMP-2 (**c**,**d**) and TIMP-3 (**e**,**f**) after 4 days treatment with 1, 10, 100 & 300 µg/mL CPX and subsequent 8-day recovery period in equine derived tendon explant cultures (n = 4). Results are expressed logarithmically. Values represent mean ± SEM and are expressed as fold change relative to control, normalised to GAPDH. ** p* < 0.05, *** p* < 0.01 vs. control.

**Table 1 genes-13-02210-t001:** Oligonucleotide Sequences of Specific Equine GOI Primers for RT-PCR.

Gene	Accession Number	Base Pair Length	Sense	Antisense
GAPDH	NM_001163856.1	1277 bp	GTGGTGAAGCAGGCATCG	AGGTGGAAGAGTGGGTGTC
MMP-3	NM_001082495.2	1802 bp	TGATGTCGGTCACTTCACTAC	AACAGCATCTCTTGGCAAATC
MMP-9	EU025852.1	2151 bp	TTGGTCCTGGCGGTCTTG	CCTGTCAGTGAGGTTAGTTAGC
MMP-13	NM_001081804.1	2727 bp	CCGTATTGATGCTGCCTATG	AACCTTCCAGAATGTCATAACC
ADAMTS-4	NM_001111299.1	2514 bp	CCCGAAATGGTGGCAAATAC	CAGTGCGGTGGTTGTAGG
ADAMTS-5	XM_023629969.1	10,009 bp	TTCCATCCTAACCAGCATTG	TCTGACCTGGGGAGTTCTTC
TIMP-1	XM_023633181.1	892 bp	GGACAACTATTGGACGAGAAG	GGATGGATGAACAGGGAAAC
TIMP-2	XM_023651899.1	3696 bp	GAGATGGAGCAGACAAGAC	TTCAGACAAGCCAGACAAG
TIMP-3	NM_001081870.2	1086 bp	GCAACTTCGTGGAGAGGTG	CGTAGCAAGGCAGGTAGTAG

## Data Availability

The data presented in this study are openly available in available in ZENODO repository at doi: 10.5281/zenodo.7341705.

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
