# Peer review of "Short-Term Exposure to Ciprofloxacin Reduces Proteoglycan Loss in Tendon Explants"

_genes, 2022, doi:10.3390/genes13122210_

Round 1

Reviewer 1 Report

This research type manuscript titled "Short-term exposure to ciprofloxacin reduces proteoglycans loss in tendon explants”.

In this study the research strategy, methods, results and discussion, but this work should follow minor revisions:

1.    The authors used a lot of times the same word “Furthermore”. I suggest change in some cases for another word. 

2.     The authors refers that Equine fetlocks were obtained from skeletally mature horses which euthanised for reasons other than musculoskeletal disorder. What are these musculoskeletal disorder? Not influence the authors studies?  

3.    Please show that GAPDH is stably expressed between all your samples and that treatment conditions have no effect on reference gene expression.    I strongly recommend to analyse qPCR data according to the geNorm guidlines using more than 1 reference gene for normalization.

4.     The discussion does not mention the limitations of the study. Please add. 

Reviewer 2 Report

This is overall a nice piece of work. It is well written and well presented. However, I have some serious concerns.

My main concern is the sample size (n=4) and lack of information about the origin of the samples – no information is provided about the gender, breed, age, or clinical presentation of the animals, and the time from euthanasia to processing of samples for example. Were all the samples processed and analysed simultaneously?

What about the controls? Were they matched to the treated samples?

It is easy to imagine a single outlier could skew the results. In that regard, I see this as a pilot study and the data have to be treated with caution and have to be explained in the right context with all the caveats (as outlined above).

The length of the PCR products (table 1) doesn’t look right. The size of the ADAMTS-5 product is apparently 10,009 bp. Usually people select small products for qPCR. Some details would be helpful. How much cDNA was used for PCR? How many cycles? What about RT reactions? Were all products analysed on one run? It is normal practice to run samples in triplicate, but there is no mention of that here.

There is no loading control (e,g, actin) on Western blots.

Most of the effects of CPX on various RNAs and proteins are only seen with non-physiological doses and are not dosage dependence, which is a concern. It is surprising the effect is very often seen at a single dose.

MMP-3 is introduced at the beginning of section 3.3 without explaining its relevance.

Although the discussion is quite thorough, it is still not easy to get a sense of the overall pathophysiology, in the context of CPX and its role in tendinopathy. Which are the important players in this pathway?

Round 2

Reviewer 2 Report

Thank you for addressing the changes.